# Influence of Nano Titanium Dioxide and Clove Oil on Chitosan–Starch Film Characteristics

**DOI:** 10.3390/polym11091418

**Published:** 2019-08-29

**Authors:** Wei Li, Kewang Zheng, Hujian Chen, Shirong Feng, Wei Wang, Caiqin Qin

**Affiliations:** 1School of Chemistry and Materials Science, Hubei Engineering University, Xiaogan 432000, China; 2Key Laboratory of Biological Resources and Environmental Biotechnology, Wuhan University, Wuhan 430000, China

**Keywords:** chitosan, starch, nano titanium dioxide, clove essential oil, edible film

## Abstract

The combined effects of nano titanium dioxide (TiO_2_-N) and clove oil (CO) on the physico-chemical, biological and structural properties of chitosan (CH)/starch (ST) films were investigated by using a solvent casting method. Results indicated that the incorporation of TiO_2_-N could improve the compactness of the film, increase the tensile strength (TS) and antioxidant activity, and decrease the water vapour permeability (WVP). As may be expected, the incorporation of CO into the film matrix decreased TS but increased the hydrophobicity as well as water vapour barrier antimicrobial and antioxidant properties. Fourier-transform infrared spectroscopy (FTIR) data supported intermolecular interactions between TiO_2_-N, CO and the film matrix. Use of a scanning electron microscope (SEM) showed that TiO_2_-N and CO were well dispersed and emulsified in the film network. Thermogravimetric (TG) and derivative thermogravimetric (DTG) curves demonstrated that TiO_2_-N and CO were well embedded in the film matrix, hence this blend film system could provide new formulation options for food packaging materials in the future.

## 1. Introduction

In the past decades, the use of synthetic packaging materials, such as polyethylene terephthalate (PET), polyethylene (PE) and polypropylene (PP), has become increasingly popular worldwide. Despite synthetic packaging materials having good physico-chemical properties and a low cost, they have caused significant environmental pollution due to their non-biodegradable and unrecyclable nature. Besides, some plastic additives, such as bisphenol A (BPA), dioctyl phthalate (DOP) and dibutyl phthalate (DBP), readily migrate from packaging films to food, thus leading to harm to human health. Therefore, the development of eco-friendly packaging materials is necessary [1,2]. Eco-friendly packaging materials usually include biodegradable films and coatings, which are based on bio-polymers (carbohydrates, proteins and lipids), additives and functional ingredients [3,4]. Among bio-polymers, chitosan and starch have been the focus of current research due to their biodegradability, good film-forming capacity, edibility and natural availability [5,6,7,8,9].

Chitosan, the product of chitin deacetylation, is a natural biopolymer which has been widely applied in the manufacturing of biodegradable films or coatings for its biocompatibility, biodegradation and good film-forming ability [10,11]. However, chitosan film has poor mechanical properties and little water resistance, which has limited its application [11]. Therefore, some strategies have been applied to modify them. For example, the use of a natural biopolymer such as starch, incorporated into chitosan film, is one of the most widespread and cheap ways used to improve film properties [12,13,14]. However, poor antioxidant, antimicrobial and water vapour barrier properties were the main disadvantages of chitosan–starch-based films. To enhance the performance thereof, several strategies have been adopted, such as incorporating inorganic nanoparticles and essential oils into the chitosan–starch-based matrix due to essential oils or inorganic nanoparticles possessing good barrier, hydrophobic, mechanical, thermal, antimicrobial and anti-oxidation properties [15,16,17,18,19,20].

Incorporation of inorganic nano-fillers, such as zinc oxide, silver, silica and clay into food packaging has been proven to enhance the physico-chemical and other properties of biodegradable films compared to the neat matrices [21,22,23]. Titanium dioxide (TiO_2_) as functional filler has been widely applied in medicines, foodstuffs and feed processing due to its non-toxicity, security, high stability and antibacterial properties [24]. It was reported that the addition of TiO_2_-N into food packaging materials could provide food prevention against growth of microorganisms [25,26]. However, TiO_2_-N offers poor anti-oxidation activity which is its main disadvantage in practice (oxidation is one of the important causes of food spoilage), whereas many plant essential oils have been proved to have excellent antioxidant and antibacterial activities [27,28]. Moreover, the essential oils were usually incorporated into the bio-polymer matrix to enhance its water-barrier properties. Clove oil (CO) is extracted from lilac and its main constituent is eugenol. CO has been reported to have excellent antimicrobial and anti-oxidation activities [29] and has received FDA approval for use within food and medicine. Recently, inorganic nanoparticles and essential oil have been proved to improve the physico-chemical and biological properties of edible film [30]. Wu et al. [31] developed an edible film based on a soy protein isolate, cinnamaldehyde and zinc oxide nanosheet. Their research showed that zinc oxide and cinnamaldehyde have a synergistic effect on the physico-chemical properties and antifungal activity of edible film. Wu et al. [32] found that a chitosan coating containing both laurel essential oil and nanosilver had better antioxidant properties than a regular coating, laurel essential oil coating and nanosilver coating.

There are, however, few reports of the application of TiO_2_-N and CO on edible films. Considering the excellent antioxidant and antibacterial activities of CO and TiO_2_-N, the aim of this work was to obtain an active film based on chitosan, starch, TiO_2_-N and CO, and investigate the influences of TiO_2_-N and CO contents on the physico-chemical, morphological, biological and structures properties of the film.

## 2. Materials and Methods

### 2.1. Materials

Chitosan (CH, *M*_v_ = 1.98 × 10^5^ g/mol, degree of deacetylation ≥ 82%) was supplied by Shandong AK Biotech Co. Ltd. (Qingdao, China). Starch (ST) was supplied by National Starch Co. Ltd. (Shanghai, China). Clove oil (CO) as a guarantee reagent, the TiO_2_-N (rutile, particle size = 40 nm, and purity ≥ 99.8%) and other reagents were all purchased from Aladdin Biochemical Co. Ltd. (Shanghai, China).

### 2.2. Film Preparation

CH solution was obtained by dissolving 1.5 g/100 mL CH in 1 mL/100 mL acetic acid with constant agitation overnight at room temperature. ST was dispersed in water to get 1.5 g/100 mL suspension, which was heated to 90 °C and held for 35 min under stirring to allow gelatinisation. The CH–ST solution was obtained by addition of 100 mL of CH solution to an equal volume of ST solution, then 18% *w*/*w* (based on CH and ST) glycerol was added as the plasticiser. Thereafter, the CH–ST solution was cooled to around 45 °C before adding TiO_2_-N and CO. To achieve better dispersion of TiO_2_-N and emulsification of CO, different concentrations (1%, 3%, 5% and 7% *w*/*w*) of TiO_2_-N was dispersed in 20 mL of distilled water and then ultrasonicated for 10 min. Different concentrations (3%, 6%, 9% and 12% *w*/*w*) of CO could be first mixed with 10% (*w*/*w*, total CO) Tween 80 and then the mixture was dissolved in 3 mL ethyl alcohol.

The film-forming dispersion (FFD) was prepared by mixing the CH–ST solution, TiO_2_-N suspension and CO emulsion. Then, the film forming solution was homogenised at 40 °C at 9000 rpm in a homogeniser for 5 min (FJ-300SH, Shanghai Specimen, Shanghai, China), filtered through cheesecloth and vacuumed. Finally, the FFD was poured into plastic discs (diameter of 90 mm) and dried at 38 °C and a relative humidity (RH) of 53% (using a saturated sodium bromide solution) to obtain the films. Before testing, all samples were stored at 25 °C and 57% RH for at least 48 h. A control film specimen was obtained in a similar manner but without any TiO_2_-N or CO.

### 2.3. Determination of Film Structure

The FTIR spectra of the sample were recorded using a Bruker Vertex 80 spectrometer equipped with an ATR accessory from 700 to 4000 cm^−1^. The thermal stability of the sample was recorded in a differential thermal analyser (STA6000, Perkins Elmer, Norwalk, CT, USA) under a nitrogen atmosphere, using a heating rate of 10 °C/min and the programmed temperature ranged from 30 °C to 800 °C. The morphology of the sample was recorded using a video recorder (D7500, Nikon, Tokyo, Japan). Microstructural analysis of the cross-sections of samples was facilitated by use of an SEM (EVO, Zeiss, Heidenheim, Germany) with an accelerating voltage of 5 kV. Before testing, the film was gold coated.

### 2.4. Physico-Chemical Properties

After using a digital micrometer (WHX, Xinzhenwei, Shanghai, China) to record the sample thickness at nine random positions, mechanical testing was performed through the method described by Zheng et al. [27]. Samples were cut into rectangular pieces (10 mm × 70 mm); the initial grip separation was 30 mm, and the crosshead displacement rate was 20 mm/min.

The moisture content (MC) of each specimen was measured by thermogravimetric analysis (TGA) [33]. The heating rate applied to the film was 5 °C/min and the programmed temperature ranged from 30 °C to 150 °C. The water contact angle was measured by a static contact-angle meter (JC2000, Powereach, Shanghai, China). Film was put on the sample stage, and the contact angle was measured immediately after dropping 1 μL distilled water on the sample surface [34].

The WVP of samples was determined by using a published method with some modifications [17]: 10 g of oven-dried calcium chloride (CaCl_2_) was poured into a permeation cup (diameter, 50 mm), and each sample (65 mm diameter) was sealed on the cup. Then the permeation cup was put in a box at a temperature of 25 °C at 78% RH and weighed every 8 h for 4 d.

Optical properties of samples were evaluated by the methods of colour difference and opacity. Opacity was measured using an UV-Vis spectrophotometer (UV1800, Jinghua, Shanghai, China) with an incident wavelength of 600 nm. The optical properties were calculated as follows:*O* = *Ab*s_600_/*L*(1)
where *L* and *Ab*s_600_ are the film thickness (mm) and absorbance at 600 nm, respectively.

The colour difference was measured by a colorimeter (WSC-C, KingKo, Shanghai, China). The values of *L**, *a** and *b** represent the intensity of black–white, red–green, and yellow–blue light, respectively. The whiteness index (WI) of the film was calculated as follows:(2)$$\mathrm{WI}=100-\sqrt{{\left(100-L^{*}\right)}^{2}+a^{*}{}^{2}+b^{*}{}^{2}}$$

### 2.5. Biological Properties

The antibacterial activity of samples was assessed using the agar diffusion method: 1 mL (10^8^ to 10^9^ CFU/mL) of microorganic liquid (*Escherichia coli or Staphylococcus aureus*) was seeded with 15 mL of agar medium. Then, samples (diameter, 6 mm) were placed on the agar plate surface, and the plate was stored at 37 °C for 24 h. The area of the inhibition zone was calculated and subtracted from the sample disk area, and this difference in area was used as the inhibition zone.

The antioxidant activity of sample was performed by the 2, 2-Diphenyl-1-pikryl-hydrazyl (DPPH) scavenging assay and 2, 2′-azino-bis(3-ethylbenzothiazoline-6-sulfonic acid (ABTS) scavenging assay. Before the test, the film (0.6 g) was soaked in 30 mL distilled water and stood for 12 h at 25 °C to obtain the film extract solution. A total of 2 mL of the DPPH methanol solution (1 mM) was mixed with 1 mL of the film extract solution, and the mixture was well shaken and put in a dark chamber at 25 to 30 °C for 1 h. The mixture was measured with a UV spectrophotometer (UV1800, Jinghua, Shanghai, China) with an incident wavelength of 517 nm. The ABTS radical scavenging activity of film was evaluated using the method of Zheng et al. with some modification [35]. Before the test, 5 mL ABTS solution (7.4 mM) and 88 μL K_2_S_2_O_8_ solution (2.6 mM) were mixed, and the mixture was stored for 12 h. A total of 2 mL ABTS working solution was mixed with 0.1 mL of film extract solution, and the absorbance was measured at 734 nm after standing for 10 min in a dark chamber. The DPPH and ABTS scavenging activities of samples were determined as follows:(3)$$\mathrm{Activity}\;(\%)=\frac{\left(A_{0}-A_{s}\right)}{A_{0}}\times 100$$
where *A*_0_ and *A*_s_ were the absorbances of the control solution and tested sample solution, respectively.

### 2.6. Statistical Analysis

Data analysis was done through SPSS software (SPSS 16.0.2, Palo Alto, CA, USA). Statistical significance of differences in means was obtained by ANOVA and the Duncan’s multiple range tests at *p* < 0.05. Data were presented as mean ± standard deviation.

## 3. Results and Discussion

### 3.1. Characterization of Film

#### 3.1.1. FTIR Spectroscopy

The IR spectra of different films are shown in Figure 1. For control specimens, a strong broad peak located at 3420 cm^−1^ was caused by the stretching vibration of free O–H, which was probably due to the presence of hydroxyl groups [36]. The bands at 1640 cm^−1^, 1558 cm^−1^ and 1410 cm^−1^ were ascribed to bound water, N–H (amide Ⅱ) stretching, and C–H bending of –CH_2_, respectively [37]. The bands appearing from 920 cm^−1^ to 1039 cm^−1^ were due to C–O–C stretching [38]. After incorporation of TiO_2_-N, partial peak intensities changed, but no noticeable wavenumber shift or peak appeared, or disappeared, which showed that only the possible formation of physical interaction occurred between the nanofiller and the biopolymer matrix. Similar results pertaining to the interaction of a bio-polymer matrix with other nanoparticles have been reported [21,39,40]. However, small modifications were found in the spectra after addition of CO. The peaks at 3420 cm^−1^, 1558 cm^−1^, 1410 cm^−1^ and 1039 cm^−1^ became sharper. Moreover, new peaks appeared at 2310 cm^−1^ and 996 cm^−1^, which were assigned to residual CO_2_ from atmosphere and C–O–C vibrations. This result indicated that incorporation of CO in the film could cause some interactions (such as hydrogen bonding or covalent bonding) between CO and the film matrix.

#### 3.1.2. Microstructure

SEM images of cross-sections of different samples are shown in Figure 2. The structure of the specimens was influenced by the presence or absence of TiO_2_-N and CO. The control film displayed a smooth, compact, continuous structure without cracks or pores. The addition of TiO_2_-N caused slight changes in the film structure, and some granules were present in the film. The number of granules was increased in the film structures with increasing TiO_2_-N concentration, however after incorporation of CO the structure became heterogeneous. As the CO content increased, the roughness of the cross-section of films was increased. The CO droplets were well embedded in the film matrix, and the number and size thereof increased with increasing CO content. The phenomenon showed the existence of CO on the structure of the film. Moreover, no evidence was observed that CO droplets were separated from the film matrix network, which indicated that CO was well emulsified and dispersed in the film matrix. The results were similar to those of Peng et al. [41].

#### 3.1.3. Thermal Stability

To evaluate the effect of TiO_2_-N and CO addition on the thermal stability of such films, TGA experiments were undertaken. The TG and DTG curves of different films are illustrated in Figure 3. Control film specimens displayed three stages in the process of thermal decomposition, ranging from 40–150 °C, 150–235 °C and 235–500 °C, ascribed to the evaporation of water and solvent, decomposition of lower molecular weight fractions, and pyrolytic decomposition of biopolymer chains, respectively. When 3% TiO_2_-N was added to control film specimens, the TG curve of this sample showed a similar behaviour to that of the control sample, which indicated that the addition of TiO_2_-N did not affect the thermal stability of the film.

As can be seen, four stages in the process of thermal decomposition were found in the film to which 3% TiO_2_-N and 9% CO had been added. The top three stages of mass loss of the film were similar to those for the control film and 3% TiO_2_-N film specimens. The fourth step took place between 380 and 460 °C and was attributed to decomposition of CO, which could be ascribed to the CO being stably embedded in the film network. Similar results were observed by Sliva [42] and Shen [43]. It was interesting to note that when 9% CO and 3% TiO_2_-N were added to the control film, the mass loss below 420 °C was lower in comparison to control and 3% TiO_2_-N films. This could be ascribed to the presence of CO, thus decreasing the moisture content of the sample. Moreover, as can be seen from the curves, the thermal degradation temperatures of the film were slightly increased after addition of 9% CO, which suggested that CO could increase the thermal stability of such films.

#### 3.1.4. Morphology Analysis

Figure 4 shows the morphology of different samples. The control film had a transparent and colourless form, however the addition of TiO_2_-N caused obvious differences in morphology: The colour changed from colourless to white and the transparency was lower. This was due to the dyeing effect of TiO_2_-N. After addition of CO, the morphology of the film underwent no significant change, letters could be observed clearly through the film, which was probably due to the emulsification and dispersion of CO in the film network.

The prepared film was flexible and glossy (Figure 4d). The morphology suggested that CO could increase the pliability of the sample, which could be attributed to interactions between the film matrix and CO. The result indicated that a continuous and flexible film material could be obtained based on CH, ST, TiO_2_-N and CO by solution casting.

### 3.2. Physico-Chemical Properties

#### 3.2.1. Mechanical Properties

To evaluate the effect of TiO_2_-N and CO addition on the mechanical properties of films, tensile tests were conducted; the values of tensile strength (TS) and elongation at break (EAB) for different films are listed in Table 1. A significant increase was observed in TS and EAB decreased upon incorporation of TiO_2_-N. These changes could be ascribed to the interfacial interaction between TiO_2_-N and bio-polymer matrix. Nanoparticles could form intermolecular hydrogen bonds or covalent bonds with the CH or ST, consequently strengthening the molecular forces between the nanoparticles and the film matrix [21,36]. It can be seen that the TS first increased rapidly when the amount of TiO_2_-N increased from 0% (33.2 MPa) to 3% (39.2 MPa), but increased little thereafter. A decrease (from 21.3% to 18.9%) in EAB film was observed upon addition of 3% TiO_2_-N, but the EAB decreased more at higher doses thereof.

As expected, the introduction of CO significantly changed the TS and EAB of the samples. The TS gradually decreased with increasing CO content: This decrease was possibly due to the weaker polymer–oil interactions partially replacing the stronger polymer–polymer interactions in the film, thus decreasing the cohesion conferred by polymer network forces. Besides, the presence of CO droplets in the film partially destroyed the continuous structure of the sample, however the EAB of samples increased with addition of CO and reached a maximum value at 30.8% when the CO content was 9%, but decreased at higher CO contents. These changes could be ascribed to the CO droplets enhancing the mobility and destroying the even structure of film (Shen et al. also found similar results [43]).

#### 3.2.2. Moisture Content and Water Contact Angle

The common methods of determining the hydrophobicity of edible film are the measurement of water contact angle and moisture uptake. Moisture content (MC) and water contact angle of samples are shown in Table 1 and Figure 5. As the amounts of TiO_2_-N and CO increased, the MC decreased and the contact angle increased. When 3% TiO_2_-N and 12% CO were added to the control film, the MC decreased from 12.5% to 8.41% and the contact angle increased from 90.5° to 114°, which indicated that the addition of TiO_2_-N and CO could effectively improve the films’ hydrophobicity and its tendency to absorb water decreased. The result was likely to have been attributable to the interactions between TiO_2_-N, CO and the film matrix reducing the tendency of hydroxyl groups to interact with water, consequently leading to a more hydrophobic matrix. Others reported that incorporation of nanoparticles or essential oil onto films results in the formation of more hydrogen bonds in the nanoparticles, oil droplets and matrix, thus enhancing the hydrophobicity of the material [19,44].

#### 3.2.3. Water Vapour Permeability

The results of WVP testing are displayed in Table 2. Significant decreases in WVP occurred after incorporation of TiO_2_-N and CO. As the content of TiO_2_-N increased, the WVP of samples decreased. When 7% TiO_2_-N was added, the WVP decreased from 1.406 × 10^−10^ to 1.288 × 10^−10^ g·m^−1^·Pa^−1^·s^−1^. The decrease in WVP could be ascribed to the nanoparticles being more water resistant than the bio-composite matrix, and infilling of these nanoparticles into the matrix could form a dense structure and may have introduced a more tortuous pathway through which the water molecules had to pass [18,19,44,45]. Likewise, the WVP was also gradually decreased as more CO was added: When 12% CO was added, the WVP decreased from 1.285 × 10^−10^ to 1.058 × 10^−10^ g·m^−1^·Pa^−1^·s^−1^, which suggested that the barrier-effect was improved. The result could be ascribed to the intermolecular interactions between CO and the film matrix, which reduced the adsorption of water in the film.

#### 3.2.4. Colour Difference and Opacity

The colour characteristic parameters and light transmittance of different films are summarised in Table 3 and Figure 6. The values of *L**, WI and opacity were significantly increased as the content of TiO_2_-N increased, which suggested that the brightness of films was affected by the addition of TiO_2_-N: This could be attributed to the brightening effect of TiO_2_-N. When CO was incorporated, *L** and WI decreased and *a** and *b** increased, which implied that the film had a lower lightness and higher red and blue shades, which was due to the surface plasma resonance characteristics of TiO_2_-N and coloured substances present in CO and its compounds. Opacity measurements indicated that the opacity was unchanged, which could be due to the CO droplets being well emulsified in the film.

### 3.3. Biological Properties

#### 3.3.1. Antimicrobial Activity

Table 2 showed the antimicrobial activities of the films. The control film showed no significant antimicrobial activity and revealed no obvious inhibition zones. This phenomenon could be ascribed to the fact that CH cannot diffuse through the adjacent agar media [46]. Many studies have indicated that there was good compatibility and hydrogen bond interaction between CH and ST molecules in the matrix [37,47]. Thus, CH molecules were supposed to be immobilised in the matrix and not diffused to generate an inhibition zone. As expected, the introduction of TiO_2_-N significantly increased the inhibition zones of the two tested bacteria and the better inhibition of the samples was found with a higher TiO_2_-N content. The effect could be ascribed to the nanoparticles having many unique effects, such as surface effects and small dimension effects, resulting in excellent biological and physico-chemical characteristics. A further increase in the size of the inhibition zones was observed when the CO was added. These results showed that the films dosed with TiO_2_-N and CO could act as an antimicrobial material. The mechanism of the action thereof against microorganisms has been investigated elsewhere: The main antibacterial mechanism of TiO_2_-N and CO was attributed to an enhanced non-specific permeability of the cytoplasm membrane, leading to the bursting of cell membranes, thus killing the cell [12,34].

#### 3.3.2. Antioxidant Activity

The DPPH radical scavenging activity of different samples is demonstrated in Figure 7 and Figure 8. The control film had little scavenging activity (only 1.62%). Likewise, when TiO_2_-N was added to the film, the scavenging activities were also under 2%, which indicated that TiO_2_-N had not imparted any antioxidant activity. However, the scavenging activity of the sample increased significantly when CO was incorporated. Figure 8 shows that samples dosed with CO could act as stronger donors of electrons or hydrogen atoms to make the purple working solution fade to yellow [16]. The scavenging activities of the film were found to be around 1.78% (without CO), with the value increasing 13.2-, 16.1-, 25.4- and 29.3-fold in presence of 3%, 6%, 9% and 12% CO, respectively. This indicated that the scavenging activity of the samples was increased upon CO addition.

The ABTS radical scavenging activity of different films is shown in Figure 9 and Figure 10. The control film had a scavenging activity of 19.46%. After the addition of TiO_2_-N, the scavenging activity decreased slightly, which could be attributed to the fact that TiO_2_-N can act as a photocatalyst in the presence of water and generate reactive oxygen species; however, CO could improve the scavenging activity. When 12% CO was incorporated to the films, the scavenging activity increased from 16.53% to 44.72%. Figure 10 shows that samples with added CO could act as stronger donors of electrons or hydrogen atoms to fade the blue colour of the reaction solutions. The result showed that CO could enhance the antioxidant ability of such films.

## 4. Conclusions

In this work, TiO_2_-N and CO were introduced to a CH/ST matrix to prepare a novel active edible film. With the addition of TiO_2_-N, the physico-chemical and antimicrobial properties of the sample increased, but the transparency decreased which suggested that the density of the film had been increased. Incorporation of CO in the films decreased TS but increased the hydrophobicity and biological properties. Results of this work indicated that a good compatibility could be obtained between TiO_2_-N, CO and the film matrix. Structural characterisation showed that the intermolecular interactions and compatibility were prepared between appropriate proportions of TiO_2_-N, CO and the film matrix. FTIR spectra showed that some interactions occurred between TiO_2_-N, CO and the film matrix. SEM and TG indicated that TiO_2_-N and CO were well-dispersed and emulsified in the film network. Active film prepared from TiO_2_-N, CO, CH and ST has good physico-chemical and biological properties, and thus offered a potential formulation option for food preservation products.

## Figures and Tables

**Figure 1 polymers-11-01418-f001:**
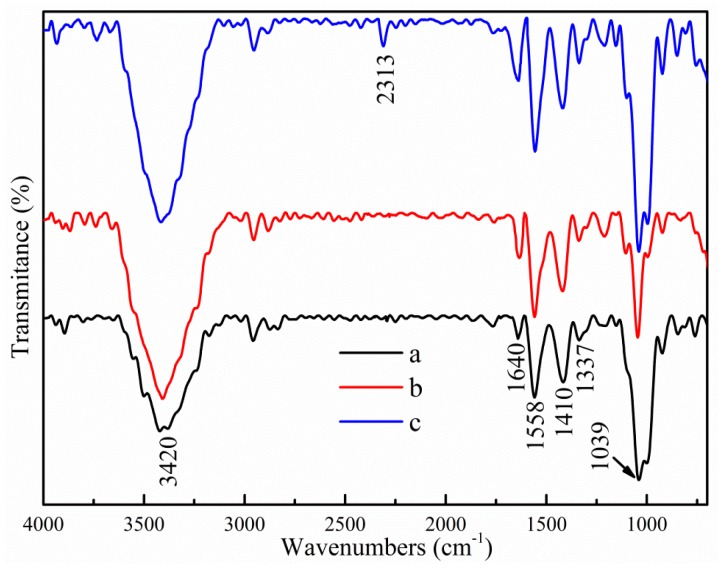
FTIR spectra of different samples. (**a**) Control; (**b**) 3% TiO_2_-N; and (**c**) 9% CO-3% TiO_2_-N.

**Figure 2 polymers-11-01418-f002:**
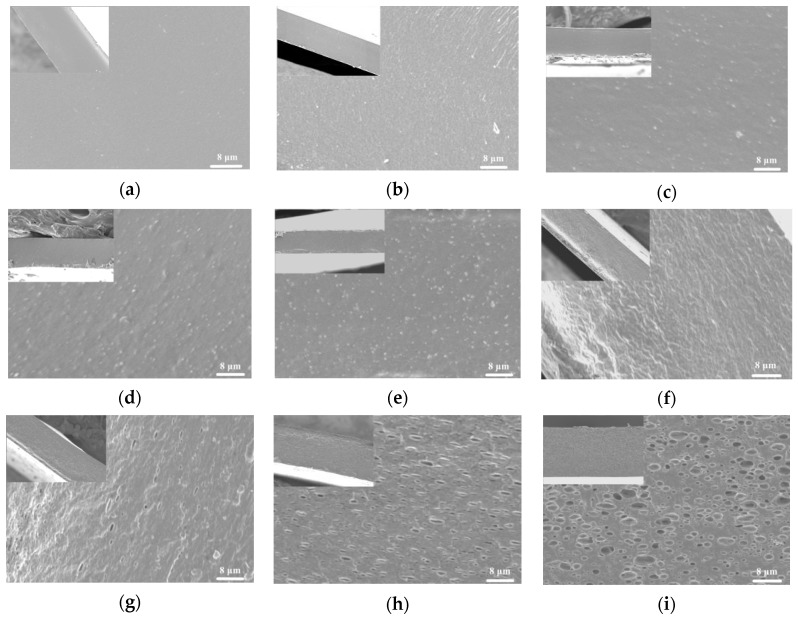
SEM micrographs of cross sections of films. (**a**) Control; (**b**) 1% TiO_2_-N; (**c**) 3% TiO_2_-N; (**d**) 5% TiO_2_-N; (**e**) 7% TiO_2_-N; (**f**) 3% CO-3% TiO_2_-N; (**g**) 6% CO-3% TiO_2_-N; (**h**) 9% CO-3% TiO_2_-N; and (**i**) 12% CO-3% TiO_2_-N.

**Figure 3 polymers-11-01418-f003:**
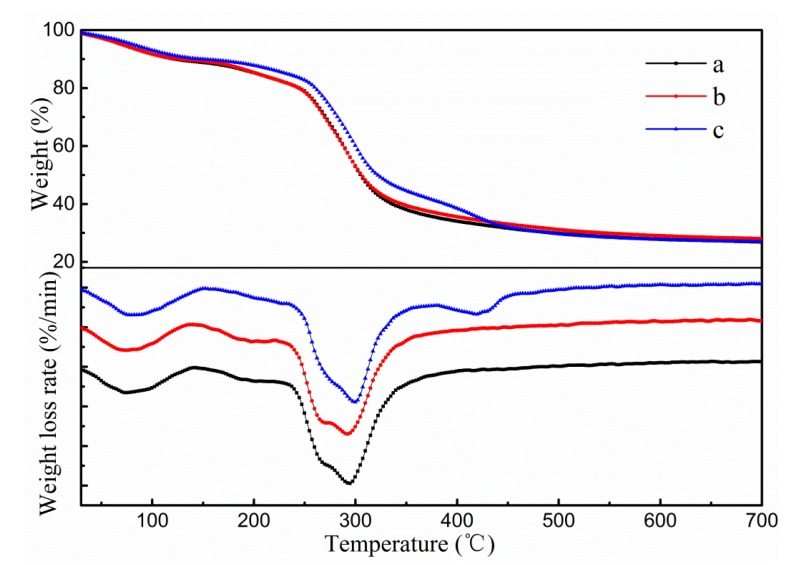
TG–DTG graphs of different samples. (**a**) Control; (**b**) 3% TiO_2_-N; and (**c**) 9% CO-3% TiO_2_-N.

**Figure 4 polymers-11-01418-f004:**
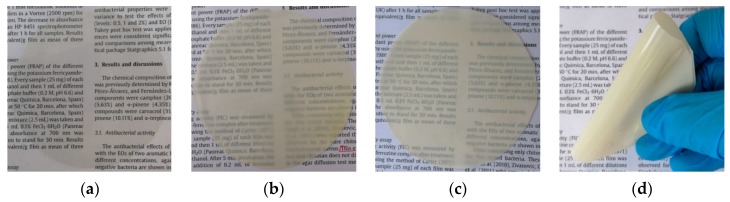
Appearance of different films. (**a**) Control; (**b**) 3% TiO_2_-N; and (**c**,**d**) 9% CO-3% TiO_2_-N.

**Figure 5 polymers-11-01418-f005:**
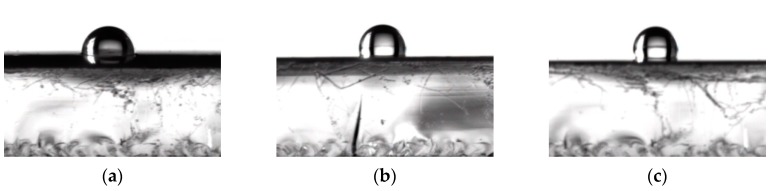
Static contact angle graphs of different films. (**a**) Control; (**b**) 3% TiO_2_-N; and (**c**) 9% CO-3% TiO_2_-N.

**Figure 6 polymers-11-01418-f006:**
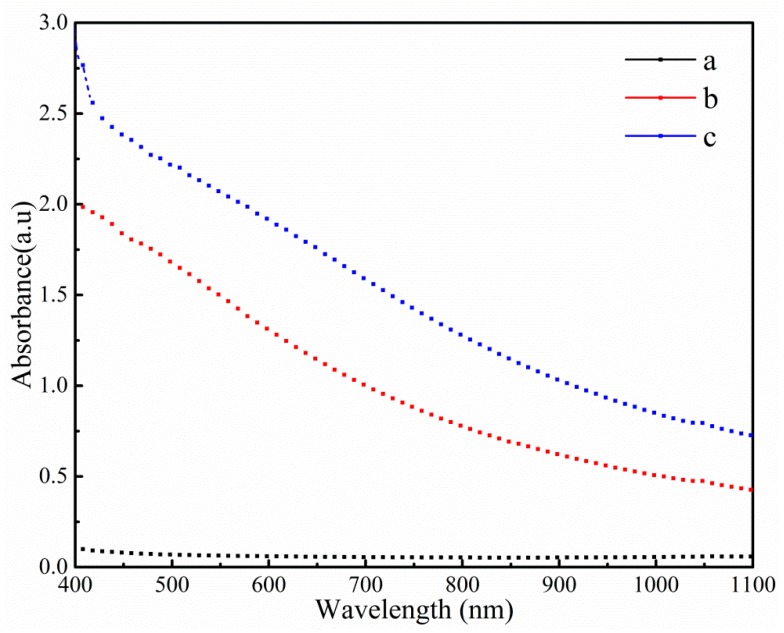
Visible absorbance for different films. (**a**) Control; (**b**) 3% TiO_2_; and (**c**) 3% TiO_2_–9% CO.

**Figure 7 polymers-11-01418-f007:**
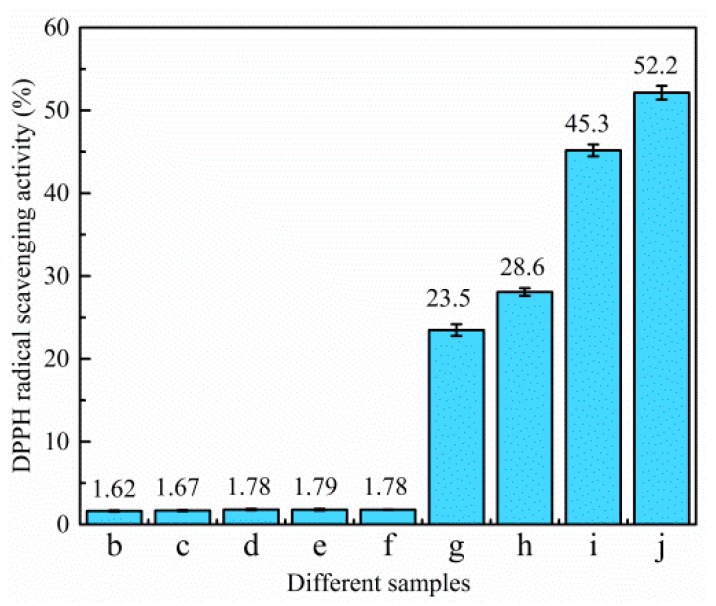
DPPH radical scavenging activity. (**b**) Control film; (**c**) 1% TiO_2_-N; (**d**) 3% TiO_2_-N; (**e**) 5% TiO_2_-N; (**f**) 7% TiO_2_-N; (**g**) 3% CO-3% TiO_2_-N; (**h**) 6% CO-3% TiO_2_-N; (**i**) 9% CO-3% TiO_2_-N; and (**j**) 12% CO-3% TiO_2_-N.

**Figure 8 polymers-11-01418-f008:**
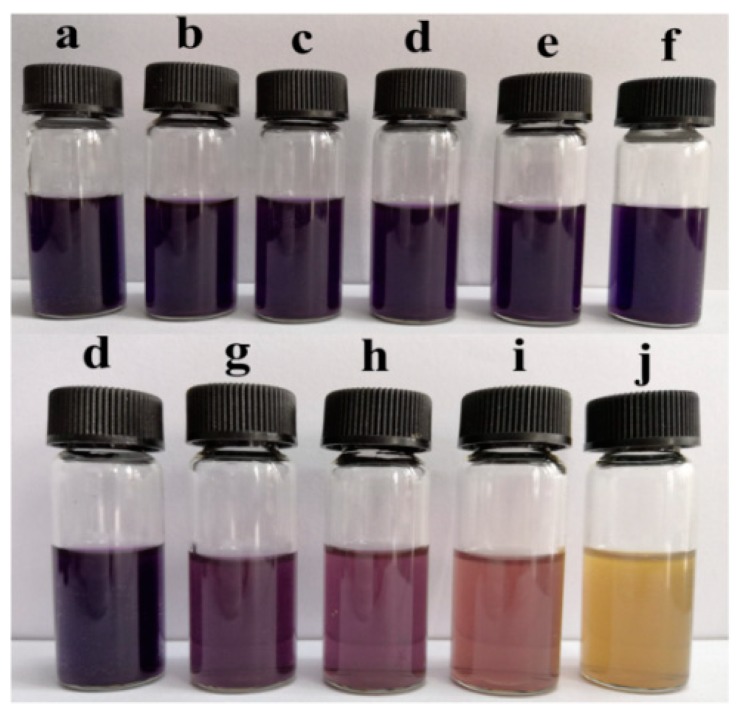
Pictures of DPPH radical scavenging activity. (**a**) DPPH methanol solution; (**b**) control film; (**c**) 1% TiO_2_-N; (**d**) 3% TiO_2_-N; (**e**) 5% TiO_2_-N; (**f**) 7% TiO_2_-N; (**g**) 3% CO-3% TiO_2_-N; (**h**) 6% CO-3% TiO_2_-N; (**i**) 9% CO-3% TiO_2_-N; and (**j**) 12% CO-3% TiO_2_-N.

**Figure 9 polymers-11-01418-f009:**
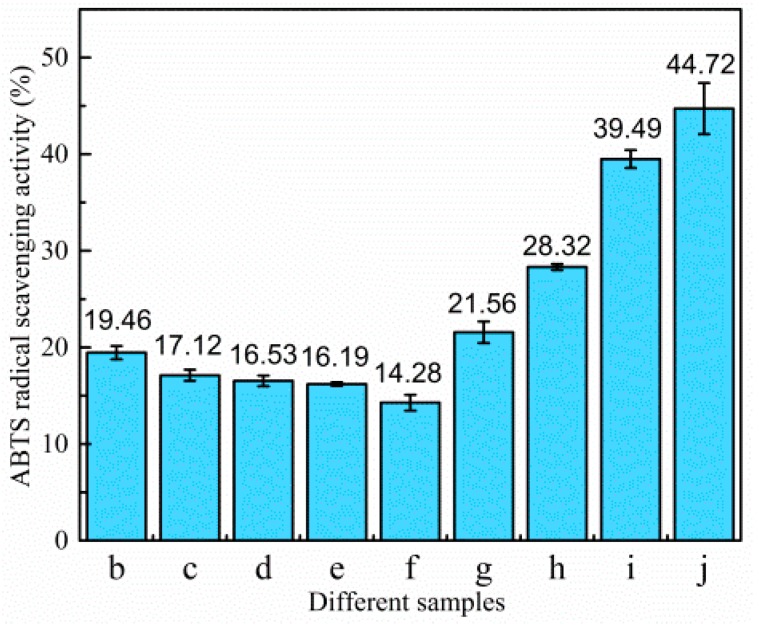
ABTS radical scavenging activity. (**b**) Control film; (**c**) 1% TiO_2_-N; (**d**) 3% TiO_2_-N; (**e**) 5% TiO_2_-N; (**f**) 7% TiO_2_-N; (**g**) 3% CO-3% TiO_2_-N; (**h**) 6% CO-3% TiO_2_-N; (**i**) 9% CO-3% TiO_2_-N; and (**j**) 12% CO-3% TiO_2_-N.

**Figure 10 polymers-11-01418-f010:**
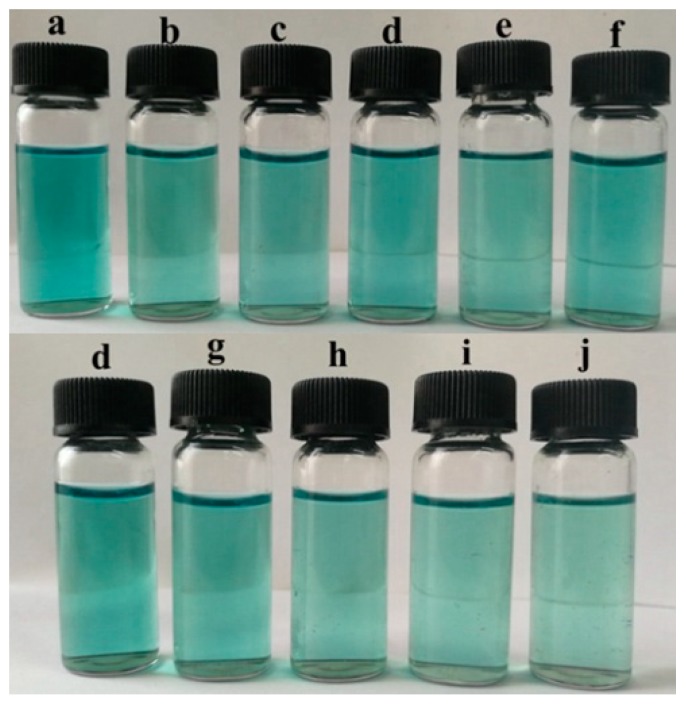
Pictures of ABTS radical scavenging activity. (**a**) ABTS solution; (**b**) control film; (**c**) 1% TiO_2_-N; (**d**) 3% TiO_2_-N; (**e**) 5% TiO_2_-N; (**f**) 7% TiO_2_-N; (**g**) 3% CO-3% TiO_2_-N; (**h**) 6% CO-3% TiO_2_-N; (**i**) 9% CO-3% TiO_2_-N; and (**j**) 12% CO-3% TiO_2_-N.

**Table 1 polymers-11-01418-t001:** Physical and mechanical properties.

Films	Thickness (mm)	TS (MPa)	EAB (%)	MC (%)	Contact Angle (°)
Control	0.065 ± 0.009 ^a^	33.2 ± 1.65 ^d^	21.3 ± 5.2 ^d^	12.5 ± 0.12 ^d^	90.5 ± 1.37 ^a^
1% TiO_2_	0.068 ± 0.007 ^ab^	37.8 ± 1.59 ^e^	20.2 ± 3.5 ^cd^	11.4 ± 0.28 ^cd^	103.2 ± 1.25 ^b^
3% TiO_2_	0.067 ± 0.007 ^ab^	39.2 ± 1.21 ^f^	18.9 ± 4.3 ^c^	10.9 ± 0.20 ^c^	106.7 ± 0.58 ^c^
5% TiO_2_	0.066 ± 0.008 ^a^	39.4 ± 0.88 ^f^	16.2 ± 3.8 ^b^	10.0 ± 0.31 ^bc^	108.8 ± 1.08 ^d^
7% TiO_2_	0.067 ± 0.008 ^ab^	37.7 ± 2.11 ^e^	13.1 ± 4.5 ^a^	9.20 ± 0.09 ^b^	109.8 ± 1.25 ^de^
3% TiO_2_–3% CO	0.068 ± 0.009 ^ab^	34.8 ± 2.21 ^de^	24.5 ± 3.8 ^e^	10.2 ± 0.11 ^bc^	108.2 ± 1.15 ^d^
3% TiO_2_–6% CO	0.070 ± 0.006 ^b^	31.3 ± 2.32 ^c^	28.9 ± 1.9 ^f^	9.62 ± 0.14 ^b^	111.4 ± 0.51 ^e^
3% TiO_2_–9% CO	0.071 ± 0.004 ^b^	29.1 ± 1.82 ^b^	30.8 ± 3.2 ^g^	8.91 ± 0.18 ^ab^	113.2 ± 0.92 ^f^
3% TiO_2_–12% CO	0.072 ± 0.004 ^b^	26.3 ± 1.14 ^a^	26.2 ± 2.8 ^ef^	8.41 ± 0.16 ^a^	114.0 ± 0.86 ^f^

Different letters in the same column indicate significant differences among formulations (*p* < 0.05).

**Table 2 polymers-11-01418-t002:** Water vapour permeability (WVP) and antimicrobial properties.

Films	WVP (10^−^^1^^0^ g/Pa·m·s)	Inhibitory Zone (mm^2^)	
		*S. aureus*	*E. coli*
Control	1.406 ± 0.008 ^a^	-	-
1% TiO_2_	1.300 ± 0.019 ^b^	30 ± 1.04 ^a^	23 ± 1.02 ^a^
3% TiO_2_	1.285 ± 0.027 ^bc^	43 ± 2.77 ^b^	29 ± 2.11 ^b^
5% TiO_2_	1.204 ± 0.014 ^c^	49 ± 3.14 ^c^	34 ± 2.15 ^c^
7% TiO_2_	1.144 ± 0.028 ^cd^	54 ± 0.89 ^d^	39 ± 1.73 ^d^
3% TiO_2_–3% CO	1.202 ± 0.121 ^c^	45 ± 1.89 ^bc^	32 ± 3.88 ^bc^
3% TiO_2_–6% CO	1.154 ± 0.098 ^cd^	48 ± 2.07 ^c^	34 ± 1.17 ^c^
3% TiO_2_–9% CO	1.123 ± 0.201 ^cd^	50 ± 3.55 ^cd^	37 ± 2.04 ^cd^
3% TiO_2_–12% CO	1.058 ± 0.109 ^d^	53 ± 3.73 ^d^	40 ± 3.21 ^d^

Different letters in the same column indicate significant differences among formulations (*p* < 0.05).

**Table 3 polymers-11-01418-t003:** Colour difference and opacity.

Films	*L**	*a**	*b**	WI	Opacity
Control	66.43 ± 0.09 ^a^	4.69 ± 0.23 ^a^	8.67 ± 0.51 ^a^	65.01 ± 0.15 ^a^	0.74 ± 0.12 ^a^
1% TiO_2_	78.29 ± 0.16 ^a^	5.30 ± 0.09 ^a^	13.86 ± 0.79 ^b^	73.70 ± 0.12 ^a^	9.02 ± 0.18 ^b^
3% TiO_2_	81.33 ± 0.24 ^a^	5.96 ± 0.24 ^a^	13.56 ± 0.84 ^bc^	76.17 ± 0.29 ^ab^	19.4 ± 0.11 ^c^
5% TiO_2_	82.36 ± 0.09 ^a^	6.14 ± 0.10 ^a^	12.65 ± 0.42 ^bc^	77.44 ± 0.31 ^b^	29.3 ± 0.24 ^d^
7% TiO_2_	83.11 ± 0.09 ^a^	6.21 ± 0.53 ^a^	12.44 ± 0.72 ^bc^	78.12 ± 0.15 ^b^	34.6 ± 0.37 ^e^
3% TiO_2_–3% CO	78.42 ±0.07 ^a^	6.98 ± 0.12 ^a^	16.47 ± 0.11 ^c^	72.51 ± 0.14 ^b^	22.4 ± 0.31 ^c^
3% TiO_2_–6% CO	77.77 ± 0.47 ^a^	7.95 ± 0.18 ^a^	17.72 ± 0.84 ^bc^	70.48 ± 0.39 ^ab^	24.4 ± 0.14 ^cd^
3% TiO_2_–9% CO	77.44 ± 0.17 ^a^	8.68 ± 0.11 ^a^	19.50 ± 0.14 ^bc^	68.94 ± 0.42 ^ab^	25.8 ± 0.22 ^cd^
3% TiO_2_–12% CO	77.24 ± 0.09 ^a^	8.87 ± 0.36 ^a^	19.52 ± 0.33 ^b^	68.73 ± 0.45 ^a^	26.9 ± 0.34 ^cd^

Different letters in the same column indicate significant differences among formulations (*p* < 0.05).

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
