# Peer review of "Influence of Nano Titanium Dioxide and Clove Oil on Chitosan–Starch Film Characteristics"

_polymers, 2019, doi:10.3390/polym11091418_

Round 1

Reviewer 1 Report

Authors explained that they prepared nanocomposite film based on chitosan and analyzed to its physiochemical, biological, and structural properties. I think of very interesting it. In addition, completeness of this article is very high, including results, discussion and composition of article, etc. If authors correct some problems to below, this article is possible to publish in 'Polymers'.

- Minor comments

Please check the typing errors of words. Please sign the special symbol accurately.  ex) 2.1 Materials : 1 line sentence 25 oC (miss) -> 25 oC (correct): Please correct the grammar of sentence generally. Please check the reference rule of 'Polymers' journal. Please replace to SEM images with high resolution of fig.2.

Author Response

Comments and Suggestions for Authors

Authors explained that they prepared nanocomposite film based on chitosan and analyzed to its physiochemical, biological, and structural properties. I think of very interesting it. In addition, completeness of this article is very high, including results, discussion and composition of article, etc. If authors correct some problems to below, this article is possible to publish in 'Polymers'.

Response: Based on your helpful comments, the language usage has been revised by a native English-speaker engaged through the auspices of a professiona proofreading service. We now believe that the language of paper should meet the standards  required for publication in  your journal and should make our work accessible to the scientific community. Both Revision using MS-Word “track changes”,  and  Language Revision  Certificate issued by the service, are all uploaded for your review reference.

1. Please check the typing errors of words. Please sign the special symbol accurately.  ex) 2.1 Materials : 1 line sentence 25 oC (miss) -> 25 oC (correct): Please correct the grammar of sentence generally.

Response: We thank this referee for this careful check and these mistakes have been corrected. We also did a thorough check ourselves and checked again by a native English speaker.

2. Please check the reference rule of 'Polymers' journal.

Response: We thank this referee for the helpful suggestion, and we have revised the it.

3. Please replace to SEM images with high resolution of fig.2. 

Response: We thank this referee for his helpful suggestions. We have replaced the SEM images with high resolution.

Reviewer 2 Report

This paper addresses the characterization of a series of chitosan/starch composites added with TiO2 nanoparticles and clove essential oil (CEO). Though the methodology used is adequate and the conclusions reached are consistent with obtained results, there are serious grammatical mistakes that still conditions its acceptance in Polymers. I do recommend a deep revision by the authors to really express what it is correct and what they want to.

In particular, and going section by section:

Abstract.

16 (TS) should be defined. L. 16 no phrase should be started with “And”. L. 17, there is no reason to find that properties modification caused by CEO are “obvious”, authors should sat “As may be expected..”. L. 19 what are “comprehensive properties”?. L. 20 “possible” and L.21 “confirmed” are unmatched concepts. It is enough to say that FTIR data support intermolecular interactions between….L. 22 and 23, I don’t understand the phrase “TG and DTG curves further confirmed TiO2-N and CEO present in the film matrix”, should it be the “presence” instead of “present”?.

Introduction

37, “good film-forming capacity, edibility and natural polymers” makes no sense to me. Should it be “good film-forming capacity, edibility and natural availability”?. L. 38, chitosan is NOT the “product of the deacetylated chitin”, it is a “deacetylated chitin”, in any case it should say “the product of chitin deacetylation”. L42 and 43, “starch” is not a film property, as it is understood from the text, “such as starch” should be moved after “Natural biopolymer”. L 44 and 45, high permeability is not a disadvantage of all biopolymer films, please, specify. L. 50, the word “nano” is excessively used. L. 56 and 57, clarify why anti-oxidation property is a main disadvantage of TiO2-N. L 63, “as” should be removed.

Materials and methods.

I do recommend a careful revision of this section. Sometimes methodology and definitions are poor or inaccurate.

81, CH and ST solutions are named but not previously defined. L. 99, a brief description about how moisture uptake has been measured should be provided. Referencing to the Nafchi method is not enough. L. 101, static contact angle (water?) is determined by a well stablished methodology, referencing to a particular method (Nafchi) makes no sense. Is there any particularity in this procedure?. L. 107, what is “Optical of the film…”, “Opacity” maybe?. L 112, L* is later referred as “lightness” in the Film color and opacity section. Definitions should be made in the Materials and methods sections. L. 113 “CIE Lab scale were present..” makes no sense to me. L. 139, “morphological” should be “morphology”?.

Results and discussion.

52, control film should be defined before being mentioned. L. 155, “And” should not start a phrase. L. 157, what is “partial peak strength”?, “peak intensity”, maybe?. L. 161, “can be seen” should be removed. Some care should be taken with the peak at 2310 cm-1, if no assignation is made it can be attributed to residual CO2 from the atmosphere. Figure has no resolution enough to follow the argumentation. A better quality set should be provided. It would be also very helpful to split the effect of TiO2-N and CEO and not simple arrange them in a 3x3 layout. L. 201, the word “respectively” should be included when assigning the TG stages. L. 205 to 211, stage assignation has been done in the previous paragraph, it makes no sense to repeat it again here. L 211 to 213 is redundant, the origin of the 380-460C stage is ascribed to CEO twice. L. 224 to 226, “morphology” is repeated and “morphological” (L. 224) should be “morphology”. L. 242 (and in many other places along the text) “experiments were tests” is wrong, experiments were done, performed, and so on…but not “tests”, in any case “tested”. L 254 to 256 should be rewritten, it is expressed exactly the opposite that it should be argued (i.e. that weaker polymer-oil interaction replaces the stronger polymer-polymer ones). This is a good example about how a deficient writing leads to a wrong interpretation. Moisture content, contact angle and water vapor permeability are justified in terms of the blockage of hydroxyls by both TiO2-N and CEO but this argumentation is reproduced in the three sections, they should be unified. L. 292, which one is the “CH-AS film”?. L. 292 “showed a gradually decreased”, L. 294, “was more water resistance”, L. 297, “gradually decreased with CEO was added” are grammatically unacceptable. L. 302, bonds with water are always hydrophilic . Table 3, “opacity” should have units considering the way it has been defined.

Conclusions

371, “compact level of the film”, is this the “compactness”?

Somewhere in the paper it is mentioned the idea of an optimization of the composite properties by the modification of the TiO2-N and CEO content. Here the TIO2-N percentage has been varied but only one TiO2-N percentage (3%) in the TiO2-N/CEO systems has been studied. With such low number of specimen I would avoid the idea of a TiO2-N/CEO composition optimization.

Author Response

Comments and Suggestions for Authors

This paper addresses the characterization of a series of chitosan/starch composites added with TiO2 nanoparticles and clove essential oil (CEO). Though the methodology used is adequate and the conclusions reached are consistent with obtained results, there are serious grammatical mistakes that still conditions its acceptance in Polymers. I do recommend a deep revision by the authors to really express what it is correct and what they want to.

Response: We thank this referee for this careful check and these mistakes have been corrected. Based on your helpful comments, the language usage has been revised by a native English-speaker engaged through the auspices of a professiona proofreading service. We now believe that the language of paper should meet the standards  required for publication in  your journal and should make our work accessible to the scientific community. Both Revision using MS-Word “track changes”,  and  Language Revision  Certificate issued by the service, are all uploaded for your review reference.

Abstract

(TS) should be defined. L. 16 no phrase should be started with “And”. L. 17, there is no reason to find that properties modification caused by CEO are “obvious”, authors should sat “As may be expected..”. 19 what are “comprehensive properties”?. L. 20 “possible” and L.21 “confirmed” are unmatched concepts. It is enough to say that FTIR data support intermolecular interactions between….L. 22 and 23, I don’t understand the phrase “TG and DTG curves further confirmed TiO2-N and CEO present in the film matrix”, should it be the “presence” instead of “present”?.

Response: We thank the referee for this helpful suggestion and we have modified these inappropriate grammar and words. The “comprehensive properties” is mainly represent the physicochemical properties. After analysis and compare in this manuscript, we would like to avoid the idea of a TiO2-N/CEO composition optimization, so we deleted it in this section.

Introduction

37, “good film-forming capacity, edibility and natural polymers” makes no sense to me. Should it be “good film-forming capacity, edibility and natural availability”?. L. 38, chitosan is NOT the “product of the deacetylated chitin”, it is a “deacetylated chitin”, in any case it should say “the product of chitin deacetylation”. L42 and 43, “starch” is not a film property, as it is understood from the text, “such as starch” should be moved after “Natural biopolymer”. L 44 and 45, high permeability is not a disadvantage of all biopolymer films, please, specify. L. 50, the word “nano” is excessively used. L. 56 and 57, clarify why anti-oxidation property is a main disadvantage of TiO2-N. L 63, “as” should be removed.

Response: We thank the referee for constructive criticisms and valuable comments, and we have revised carefully this section.

Materials and methods.

I do recommend a careful revision of this section. Sometimes methodology and definitions are poor or inaccurate.

81, CH and ST solutions are named but not previously defined.

Response: We thank the referee for valuable comments, and we have defined CH and ST in Abstract.

 99, a brief description about how moisture uptake has been measured should be provided. Referencing to the Nafchi method is not enough. L. 101, static contact angle (water?) is determined by a well stablished methodology, referencing to a particular method (Nafchi) makes no sense. Is there any particularity in this procedure?.

Response:  We thank the referee for constructive criticisms and valuable comments, and we have redescribed these experimental methods. “The moisture content (MC) of each specimen was measured by thermogravimetric analysis (TGA) [33]. The heating rate applied to the film was 5 °C/ min and the programmed temperature ranged from 30 °C to 150 °C. The water contact angle was measured by a static contact-angle meter (JC2000, Powereach, China). Film was put on the sample stage, the contact angle was measured immediately after dropping 1 μL distilled water on the sample surface [34].”

107, what is “Optical of the film…”, “Opacity” maybe?. L 112, L* is later referred as “lightness” in the Film color and opacity section. Definitions should be made in the Materials and methods sections. L. 113 “CIE Lab scale were present..” makes no sense to me. L. 139, “morphological” should be “morphology”?.

Response: We thank the referee for valuable comments, and we have revised carefully this section.  “Optical properties of samples were evaluated by the methods of colour difference and opacity. Opacity was measured using an UV-Vis spectrophotometer (UV1800, Jinghua, China) with an incident wavelength of 600 nm. The optical properties were calculated as follows:

O = Abs600/L

Where L and Abs600 are the film thickness (mm) and absorbance at 600 nm, respectively.

The colour difference was measured by a colorimeter (WSC-C, KingKo, China). The values of L*, a*, and b* represent the intensity of black-white, red-green, and yellow-blue light, respectively. The whiteness index (WI) of the film was calculated as follows”

Results and discussion.

52, control film should be defined before being mentioned.

Response: We have defined it in Line 102. A control film specimen was obtained in a similar manner but without any TiO2-N or CO.”

155, “And” should not start a phrase. L. 157, what is “partial peak strength”?, “peak intensity”, maybe?. L. 161, “can be seen” should be removed.

Response: We thank the referee for valuable comments, and these mistakes have been corrected

 Some care should be taken with the peak at 2310 cm-1, if no assignation is made it can be attributed to residual CO2 from the atmosphere.

Response: We thank the referee for valuable comments, and we have revised it. Moreover, new peaks appeared at 2310 cm-1 and 996 cm-1, which were assigned to residual CO2 from atmosphere and C-O-C vibrations.

Figure has no resolution enough to follow the argumentation. A better quality set should be provided. It would be also very helpful to split the effect of TiO2-N and CEO and not simple arrange them in a 3x3 layout.

Response: We thank this referee for his helpful suggestions. We have revised it and replaced the SEM images with high resolution.

201, the word “respectively” should be included when assigning the TG stages.

Response: We thank the referee for valuable comments, and the mistake have been corrected.

205 to 211, stage assignation has been done in the previous paragraph, it makes no sense to repeat it again here. L 211 to 213 is redundant, the origin of the 380-460C stage is ascribed to CEO twice.

Response: We thank the referee for valuable comments, and we have revised it. As can be seen, four stages in the process of thermal decomposition were found in the film to which 3% TiO2-N and 9% CO had been added. The top three stages of mass loss of the film were similar to those for the control film and 3%TiO2-N film specimens. The fourth step took place between 380 and 460 and was attributed to decomposition of CO, which could be ascribed to the CO being stably embedded in the film network.

224 to 226, “morphology” is repeated and “morphological” (L. 224) should be “morphology”. L. 242 (and in many other places along the text) “experiments were tests” is wrong, experiments were done, performed, and so on…but not “tests”, in any case “tested”.

Response: We thank this referee for his careful check and these mistakes have been corrected.

L 254 to 256 should be rewritten, it is expressed exactly the opposite that it should be argued (i.e. that weaker polymer-oil interaction replaces the stronger polymer-polymer ones). This is a good example about how a deficient writing leads to a wrong interpretation.

Response: We thank this referee for his careful check and these mistakes have been corrected.

Moisture content, contact angle and water vapor permeability are justified in terms of the blockage of hydroxyls by both TiO2-N and CEO but this argumentation is reproduced in the three sections, they should be unified.

Response: We thank the referee for this helpful suggestion. We merged the sections of moisture content and contact angle together. But water vapor permeability was still as a single section. The common methods of determining the hydrophobicity of edible film are the measurement of water contact angle and moisture uptake. Moisture content (MC) and water contact angle of samples are shown in Table 1 and Fig. 3. As the amounts of TiO2-N and CO increased, the MC was decreased and the contact angle increased. When 3% TiO2-N and 12% CO were added to the control film, the MC decreased from 12.5% to 8.41% and the contact angle increased from 90.5° to 114°, which indicated that the addition of TiO2-N and CO could effectively improve the films’ hydrophobicity and its tendency to absorb water decreased. The result was likely to have been attributable to the interactions between TiO2-N, CO and the film matrix reducing the tendency of hydroxyl groups to interact with water, consequently leading to a more hydrophobic matrix. Others reported that incorporation of nano-particles or essential oil onto films results in the formation of more hydrogen bonds in the nano-particles, oil droplets, and matrix, thus enhancing the hydrophobicity of the material [19, 44].

292, which one is the “CH-AS film”?. L. 292 “showed a gradually decreased”, L. 294, “was more water resistance”, L. 297, “gradually decreased with CEO was added” are grammatically unacceptable. L. 302, bonds with water are always hydrophilic .

Response: We thank this referee for his careful check and these mistakes have been corrected.

Table 3, “opacity” should have units considering the way it has been defined.

Response: We thank the referee for this helpful suggestion. The unit of opacity is 1, and it was usually omitted. It has been reported by other literatures. (Rambabu K. et al. International Journal of Biological Macromolecules. 2019, 126, 1234-1243. Pérez-Córdoba L. J. et al. Food Hydrocolloids, 2018, 79, 544-559)

Conclusions

371, “compact level of the film”, is this the “compactness”?

Response: We thank this referee for his careful check and these mistakes have been corrected.

Somewhere in the paper it is mentioned the idea of an optimization of the composite properties by the modification of the TiO2-N and CEO content. Here the TIO2-N percentage has been varied but only one TiO2-N percentage (3%) in the TiO2-N/CEO systems has been studied. With such low number of specimen I would avoid the idea of a TiO2-N/CEO composition optimization.

Response: We thank the referee for this helpful suggestion. After analysis and compare, we have deleted it in the manuscript.

Reviewer 3 Report

The manuscript describes enhancement of chitosan / starch composite film characteristics following addition of inorganic nanomaterials and essential oils. Such approach was already demonstrated for others edible films. The novelty is related to the combination of chitosan (CH)/starch (ST) edible film with nano titanium dioxide (TiO2-N) and clove essential oil (CEO). The manuscript needs modifications before publication.

The title of the manuscript is too generic and do not describe the content. A title like Influence of the nano-titanium dioxide and clove essential oil on chitosan -starch film characteristics could better reflect the content of the manuscript. On Introduction Section it is necessary to add, after line 62, information about the combined effects of the inorganic nanomaterials and essential oil on active film characteristics. Several recent papers on this field are suggested: Wu et al. Colloids and Surfaces B: Biointerfaces 180 (2019): 31-38; Wu et al. Food chemistry295 (2019): 16-25; Aguilar-Sánchez et al. Molecules12 (2019): 2340. Line 28, acronyms shall be used only after using the entire name of the product. Therefor shall be polypropylene (PP), polyethylene (PE) and polyethylene terephthalate (PET), Line 38, “nature biological polysaccharide” is rather a pleonasm and it is not totally correct – biobased polysaccharides is more exact – authors just mentioned before that chitosan is produced by chitin deacetylation. Therefore, is not a natural product – is produced based on a natural product, chitin. On Material and method section, the presentation of the method shall follow the order from Results and discussion section, respectively the morpho-structural characteristics, than physico-chemical characteristics and on the end, the antioxidant activity and the anti-bacterial activity. The methods mentioned on Lines 98,100 and 1001 shall be briefly described, according to instructions to authors. At least one concentration of clove oil introduced alone into the film matrix shall be tested for the effects on antioxidant properties – preferably 6%. Titanium oxide could influence the antioxidant activity of the clove oil. It is well-known that titanium oxide act as a photocatalyst in the presence of water and generate reactive oxygen species - Nosaka  & Nosaka (2017) Chemical reviews117(17), 11302-11336. The presence of reactive oxygen species could either decrease the antioxidant activity of clove oil (by oxidation), either enhance the antioxidant activity – by dimerization of the eugenol, the main component of clove oil, to the more antioxidant dimeric form – Ogata et al., 2000,  Chemical and Pharmaceutical Bulletin10 (2000): 1467-1469. This potential interaction shall be investigated by testing at least one concentration of clove oil. The antioxidant activity shall be removed from biological properties. The biological relevance of the “antioxidant activity” determined by a chemical reactions in a test tube is limited, as was recently  argumented by a team of international experts – Granato et al (2018). Food chemistry264, 471-475. The chemical estimated “antioxidant activity” is just a screening method. Additional antioxidant tests, at least the improved ABTS radical cation decolorization assay, shall be done in order to substantiate the effects generated by clove oil addition on the film properties. The presentation of the name of the company producing the equipment shall be used, not only the commercial name of the equipment and the city where the company is located. On Results and discussion section the tables should be placed as close as possible to main text where they are first time cited. Example – table 2 shall start on Line 291 and not on Line 305.

Author Response

Comments and Suggestions for Authors
The manuscript describes enhancement of chitosan / starch composite film characteristics following addition of inorganic nanomaterials and essential oils. Such approach was already demonstrated for others edible films. The novelty is related to the combination of chitosan (CH)/starch (ST) edible film with nano titanium dioxide (TiO2-N) and clove essential oil (CEO). The manuscript needs modifications before publication.

Response: We thank this referee for his careful check and these mistakes have been corrected. Based on your helpful comments, the language usage has been revised by a native English-speaker engaged through the auspices of a professiona proofreading service. We now believe that the language of paper should meet the standards  required for publication in  your journal and should make our work accessible to the scientific community. Both Revision using MS-Word “track changes”,  and  Language Revision  Certificate issued by the service, are all uploaded for your review reference.

The title of the manuscript is too generic and do not describe the content. A title like Influence of the nano-titanium dioxide and clove essential oil on chitosan -starch film characteristics could better reflect the content of the manuscript.

Response: We thank the referee for valuable suggestions, and we have revised the title of the manuscript. Title: “Influence of nano titanium dioxide and clove oil on chitosan-starch film characteristics”

On Introduction Section it is necessary to add, after line 62, information about the combined effects of the inorganic nanomaterials and essential oil on active film characteristics. Several recent papers on this field are suggested: Wu et al. Colloids and Surfaces B: Biointerfaces 180 (2019): 31-38; Wu et al. Food chemistry295 (2019): 16-25; Aguilar-Sánchez et al. Molecules12 (2019): 2340.

Response: We thank this referee for his careful check and helpful suggestions. We have added the information in the introduction section. “Recently, inorganic nanoparticles and essential oil have been proved to improve the physico-chemical and biological properties of edible film [30]. Wu et al. [31] developed an edible film based on soy protein isolate, cinnamaldehyde and zinc oxide nanosheet. Their research showed that zinc oxide and cinnamaldehyde have a synergistic effect on the physico-chemical properties and antifungal activity of edible film. Wu et al. [32] found that chitosan coating containing both laurel essential oil and nanosilver had better antioxidant properties than regular coating, laurel essential oil coating, and nanosilver coating.”

Line 28, acronyms shall be used only after using the entire name of the product. Therefor shall be polypropylene (PP), polyethylene (PE) and polyethylene terephthalate (PET),

Response: We thank the referee for valuable comments, and we have revised it.

Line 38, “nature biological polysaccharide” is rather a pleonasm and it is not totally correct – biobased polysaccharides is more exact – authors just mentioned before that chitosan is produced by chitin deacetylation. Therefore, is not a natural product – is produced based on a natural product, chitin.

Response: We thank this referee for his careful check and these mistakes have been corrected. “Chitosan, the product of chitin deacetylation, is a nature biopolymer which has been widely applied to the manufacture of biodegradable films or coatings for its biocompatibility, biodegradation and good film forming ability”

 On Material and method section, the presentation of the method shall follow the order from Results and discussion section, respectively the morpho-structural characteristics, than physico-chemical characteristics and on the end, the antioxidant activity and the anti-bacterial activity.

Response: We thank this referee for the helpful suggestion, and we have revised them.

The methods mentioned on Lines 98,100 and 1001 shall be briefly described, according to instructions to authors.

Response: We thank this referee for the helpful suggestion, and we have redescribed these experimental methods. “The moisture content (MC) of each specimen was measured by thermogravimetric analysis (TGA) [33]. The heating rate applied to the film was 5 °C/ min and the programmed temperature ranged from 30 °C to 150 °C. The water contact angle was measured by a static contact-angle meter (JC2000, Powereach, China). Film was put on the sample stage, the contact angle was measured immediately after dropping 1 μL distilled water on the sample surface [34].”

At least one concentration of clove oil introduced alone into the film matrix shall be tested for the effects on antioxidant properties – preferably 6%. Titanium oxide could influence the antioxidant activity of the clove oil. It is well-known that titanium oxide act as a photocatalyst in the presence of water and generate reactive oxygen species - Nosaka  & Nosaka (2017) Chemical reviews, 117(17), 11302-11336. The presence of reactive oxygen species could either decrease the antioxidant activity of clove oil (by oxidation), either enhance the antioxidant activity – by dimerization of the eugenol, the main component of clove oil, to the more antioxidant dimeric form – Ogata et al., 2000, Chemical and Pharmaceutical Bulletin10 (2000): 1467-1469. This potential interaction shall be investigated by testing at least one concentration of clove oil.

Response: We thank the referee for constructive criticisms and valuable comments. To investigate that, we conducted the additional experiments. In the additional experiments, we found that 6%CO-3%TiO2-N film had a better Antioxidant activity than 6%CO film. The experimental data showed that 6% CO film has the radical scavenging activity of DPPH and ABTS about 25.9% and 35.2% ,respectively. The 6%CO-3%TiO2-N film has the the radical scavenging activity of DPPH and ABTS about 28.1% and 39.3% ,respectively. These results indicated that the antioxidant activity of clove oil could be enhanced by dimerization of the eugenol.

The antioxidant activity shall be removed from biological properties. The biological relevance of the “antioxidant activity” determined by a chemical reactions in a test tube is limited, as was recently  argumented by a team of international experts – Granato et al (2018). Food chemistry, 264, 471-475. The chemical estimated “antioxidant activity” is just a screening method. Additional antioxidant tests, at least the improved ABTS radical cation decolorization assay, shall be done in order to substantiate the effects generated by clove oil addition on the film properties.

Response: Thanks the referee for constructive criticisms and valuable comments, we expanded the ABTS scavenging assay in the manuscript.

The presentation of the name of the company producing the equipment shall be used, not only the commercial name of the equipment and the city where the company is located.

Response: We thank this referee for the helpful suggestion, and we have revised the it.

Round 2

Reviewer 2 Report

I have revised the new version of the manuscript and most of my recommendations have been taken into consideration. With these and other modifications performed, I think the paper is now suitable for publication in Polymers. Still, minor grammatical corrections should be made by I consider them within the standard editing activity of the journal. They do not affect the science and the argumentation in the text. However, there is only one additional recommendation, the number of significant figures in numerical data in table 3 should be modified, particularly those of TS, EAB and contact angle values. With error around 2, 3-4 and 1, respectively, it is wrong to provide 1 or two decimals. With this minor correction everything will be OK and no further revision would be necessary. 

Reviewer 3 Report

The manuscript was significantly improved, all my observation was addressed by the major revision done to this manuscript.